# *Listeria monocytogenes* and *Listeria ivanovii* Virulence and Adaptations Associated with Leafy Vegetables from Small-Scale Farm and a Shift of Microbiota to a New Niche at Markets: A Systematic Review

**DOI:** 10.3390/microorganisms14010076

**Published:** 2025-12-29

**Authors:** Dineo Attela Mohapi, Sebolelo Jane Nkhebenyane

**Affiliations:** Department of Life Science, Centre for Applied Food Safety and Biotechnology, Central University of Technology, Free State, Private Bag X20539, Bloemfontein 9301, South Africa; snkheben@cut.ac.za

**Keywords:** *Listeria*, serotypes, leafy greens, survival strategies, virulence factors, adaptation

## Abstract

The study conducted a review of *Listeria* prevalence, virulence, and adaptations associated with leafy vegetables from small-scale farms and their journey to markets. PubMed, Taylor and Francis, Oxford, and Google Scholar databases were utilised to search for English-language journal articles published between January 1992 and 2025. Studies utilised multi-locus sequence typing (MLST), polymerase chain reaction–restriction fragment length polymorphism (PCR-RFLP), multiplex PCR, pulsed-field gel electrophoresis (PFGE), and whole genome sequencing WGS, confocal scanning laser microscopy technique for the detection of *Listeria* species, followed by transcriptomic, phenotypic analyses, strand-specific RNA-sequencing, and membrane lipid profiling. ST_5_, ST_121_, and ST_321_ are considered predominant and virulent and have been identified in two ready-to-eat commodities, while ST_1_, ST_2_, and ST_204_ are considered hypervirulent strains in food processing environments. Immunocompromised groups can experience severe life-threatening infections, even death. Significant economic losses due to shutdowns for sanitary procedures can occur, impacting food security.

## 1. Introduction

The genus *Listeria* comprises these species: *L. monocytogenes*, *L. grayi*, *L. innocua*, *L. ivanovii*, *L. seeligeri*, and *L. welshimeri.* Only two *Listeria* pathogens, *L. monocytogenes* and *L. ivanovii*, are considered pathogenic for humans and animals [1]. The species *Listeria ivanovii* (*L. ivanovii*) and *Listeria monocytogenes* (*L. monocytogenes*) are considered pathogenic and share many virulence factors, strategies, and mechanisms of pathogenicity [2]. *Listeria* toxins and virulent factors cause illness such as gastroenteritis in infants, healthy individuals, pregnant women, the elderly, and immunocompromised individuals, as well as the deadly disease, Listeriosis, leading to septicaemia and meningitis, spontaneous abortion or damage to the foetus in pregnant women [3]. High prevalence of untreated Human Immunodeficiency Virus infections, endemic anaemia, high fertility rate, and a gradually increasing proportion of elderly persons expand the fraction of listeriosis-susceptible groups among African populations. As precautionary measures, African countries should implement systems for the detection and reporting of listeriosis cases, and food safety regulations that set *L. monocytogenes* standards and limits for high-risk ready-to-eat foods [4]. Food contaminated with fewer *L. monocytogenes* may also infrequently cause febrile illness, termed non-invasive gastrointestinal disease. Toxins are toxic substances within a bacterium that confer pathogenic traits, depending on the type of toxin. When ingested through contaminated food, they manipulate the human immune system, resulting in gastrointestinal infection and other severe complications, including environmental toxicology [5]. *Listeria monocytogenes* can thrive and survive under a wide range of growth conditions during food processing, replicate at refrigeration temperatures during storage, and spread to other batches of food [6]. It is well able to tolerate salt conditions up to 20% and a broad pH range between 4.6 and 9.5, as well as relatively low water activity (aW < 0.90) [6]. The temperature-replicating ability enhances survival characteristics, such as hydrophilicity, and induces biofilm formation in response to cold temperatures, thereby increasing attachment and resistance to detergents in many food industries [7]. Studies highlighted that *L. monocytogenes* motility is temperature dependent and the bacterium is non-motile at high temperature (37 °C) and highly motile at low temperatures, about 28 °C and below [7]. Efflux pump *bcrABC* cassette, *Ide*, *mdrL*, *qacH*, *qacA*, *qacE*11-*sul*, and *emrE* are associated with resistance to several disinfectants, including the quaternary ammonium compounds and biocides [8]. *Listeria monocytogenes* is an opportunistic foodborne intracellular pathogen that can cause illness [9] with a significant mortality rate of 20–30% [10].

Persistence of *L. monocytogenes* and *L. ivanovii* strains is contributed to by extrinsic and intrinsic factors, including poor hygiene, ineffective sanitisers, and the presence of specific genes responsible for biofilm formation and resistance [11]. Bacterial virulence is a multifactorial trait, and genes are involved in various phases of the infectious process, including invasion, intracellular multiplication, and spreading [12]. *Listeria monocytogenes* can infect a wide range of animal species, including birds and humans, whereas *L. ivanovii*, formerly known as *L. monocytogenes* serotype 5-ST5, is also considered an emerging pathogen and can infect both livestock and humans [13,14]. It is imperative to note that pathogen *L. ivanovii* is most known to only infect small ruminants, including cattle (*Bos Taurus*), and these livestock act as reservoirs [15]. Recently, *L. ivanovii* specific sequences are being detected in soil, human and animal hosts including sludge and sediments, and *L. ivanovii is* reported to be the second abundant species in humans, specifically in the gut and skin [13,14], which it can adhere to and penetrate human amnion-derived cells, lysing the phagosomal membrane, polymerising and reorganising the host cell actin in the form of tails, and spreading from cell to cell. It was concluded that *L. ivanovii*, like *L. monocytogenes*, is considered a facultative intracellular pathogen that utilises specific and unique traits and strategies to infect cells and tissues. Infections are not common in healthy individuals, but the high spike in mortality rates and co-morbidities is associated with infection [16]. Another study reported that 2–6% of healthy people had *L. monocytogenes* in faecal samples, concluding that a patient’s health status significantly impacts *Listeria* infection [17].

*Listeria monocytogenes* has been associated with meats, seafood, and dairy products; however, fresh fruits and vegetables, such as pre-cut and frozen fruit, leafy greens, sandwiches, and wraps, are now being linked to *L. monocytogenes* [18]. Various *Listeria* infection strains, including virulence factors, are characterised by the lineage commonly found in various isolates during outbreaks. Lineage I, such as serotypes 1/2b, 3b, 4b, 4d, 4e, and 7 of *L. monocytogenes*, is usually associated with human listeriosis and is thus considered more pathogenic than other lineage groups [19]. Lineage II (serotypes 1/2a, 1/2c, 3a, and 3c) *L. monocytogenes* is commonly recovered from foods, food-processing facilities, and natural or agricultural environments and is reported to be able to grow under and thrive under stress conditions [2,16]. Lineage II strains are considered more prone to be linked to food, while lineage I strains are associated with clinical isolates from listeriosis outbreaks [20]. Lineage III and IV strains are relatively rare and usually associated with animal infections [21].

Both *L. monocytogenes* and *L. ivanovii* are characterised as facultative intracellular pathogens due to their ability to cross the intestinal barrier, proliferate within macrophages, epithelial cells, and endothelial cells, and ultimately induce cell-to-cell spread, causing infection [15]. Most of the genes—the virulence gene cluster responsible for the various steps of the intracellular infectious process—are grouped in a central virulence gene cluster on the chromosomes of *L. ivanovii* and *L. monocytogenes*, whose transcription is positively regulated by *PrfA*, the main pathogenicity island [13,22]. *Listeria* pathogenicity island LIPI-1 harbours the main virulence genes, including *prfA*, *plcA*, *hly*, *mpl*, *actA*, and *plcB*, which are regulated by the transcriptional regulator *PrfA* [23]. The most important virulence determinants, including *prfA*, *plcA*, *hly*, *mpl*, *actA*, and *plcB*, are clustered together to comprise LIPI-1. LIPI-2 (*inlA*, *inlB*, *inlC*, *and inlJ*), LIPI-3 (*llsA*, *llsG*, *llsH*, *llsX*, *llsB*, *llsY*, *llsD*, and *llsP*) [24], and LIPI-4, which includes *licG*, *licB*, *licA,* and have also been identified as carrying vital virulence factors [25,26]. Each species has a major virulence determinant: secretory pore-forming proteins, including ivanolysin O and listeriolysin O, the i/o gene product of *L. ivanovii* and the *hly* gene product of *L. monocytogenes*, respectively. Additionally, in *L. ivanovii*, the structural gene, *smcl*, encodes a potent sphingomyelinase C and is also responsible for bacterial escape from phagosomes. LIPI-3 is mostly associated with lineage I and contains the *llsX* gene, which encodes listeriolysin S (LLS), a haemolytic and cytotoxic factor that contributes to pathogenicity [27]. The presence of virulence factors such as O (LLO encoded by *hlyA*), actin (*actA* internalins encoded by *inlA*, *inlC*, *inlJ*), listeriolysin, *iap* (invasion-associated protein encoded by *iap*), phosphatidyl-inositol-phospholipase C (PI-PLC encoded by *plcA*), including virulence regulator (encoded by *prfA*), in *Listeria monocytogenes* and some in *Listeria ivanovii* significantly regulates their pathogenicity.

Previous studies have reported that lineage I hypervirulent strains, which are predominant in Western countries, such as CC_1_, CC_2_, CC_4_, and CC_6_, are correlated with listeriosis clinical infections. In addition, the hypovirulent clones, such as CC_8_, CC_9_, CC_101_, CC_121_, and CC_204_, under lineage II, were associated with food contamination [28,29]. Surface contamination of meat and vegetables is relatively common, with up to 15% of these foods harbouring the organism [30]. Human listeriosis caused by *L. ivanovii* is considered rare, and the pathogen has few cases reported in the scientific literature [31,32], apart from those that have been investigated and reviewed [1]. Conversely, *L. ivanovii* is endowed with a complex system of virulence factors, constituting an evolving pathogenic potential that should be carefully studied and monitored [33].

Despite irrigation water sources and fertilisers contributing to microbial density, socioeconomic parameters such as gender and education may influence farmer hygiene practices, suggesting that these traits should be considered when planning farmer support interventions [34]. Significant impacts of Listeriosis are recorded, including mortality of 204 confirmed cases, with infants having the highest percentage of fatalities (42%), at an estimated cost of over USD 260 million, as well as the hospitalisation of individuals with one-month recovery at an estimated cost of USD 10.4 million, and productivity losses estimated at USD 15 million in South Africa [35]. Based on data generated during the listeriosis outbreak, *L. monocytogenes* was added to the South African list of mandatory notifiable medical diseases, and surveillance systems in the food chain were strengthened to assist in preventing and facilitating early detection of the infection [36]. However, various studies identified the presence of *L. monocytogenes* in South African fruits and leafy vegetables as vehicles, their multidrug resistance as an immediate danger for the health security of the public, the re-enforcement of food safety surveillance and emergency responses for consumers and farming operations to ensure food safety for consumers as required by the Sustainable Development Goals to mitigate produce contamination that could result in a health crisis [37,38,39,40]. A combination of responsibility from all levels of stakeholders in the food retail/markets sector is needed to improve food safety and prevent breaches, as there is still a gap. Underreporting, particularly from primary healthcare services, contributes to poor case reporting of listeriosis in South Africa, creating a gap that can be filled through cooperation and engagement from both public and private healthcare services. In addition, strong governance of the food safety system is required to enable effective legislation and enforcement [41] and understanding who the main food safety governance actors are, their roles, responsibilities and linkages, including leveraging of resources, with power relations considered to locally contextualised solutions to address food safety [42]. There is still a gap in South Africa regarding mandatory microbial limits, particularly for leafy vegetables, and a need for a strengthened surveillance system on small-scale farms. South Africa has strict standards for several commodities and limits on microbial counts. However, there is no set limit for *L. monocytogenes* and *L. ivanovii* found in leafy green vegetables. Hence, the rule of thumb in such cases is that leafy green vegetables should not contain any pathogenic *Listeria* and should not be detected in ready-to-eat foods, while following a zero-tolerance policy for ready-to-eat foods. Strong regulatory governance for the food safety system is essential, as it will compel the enforcement of food safety regulations and standards. Fresh vegetables, small-scale farms, and markets are imperative production and supply sectors, as many consumers depend on their products for their daily nutrition.

A critical measure for improving food safety against these pathogens is to develop mechanisms, knowledge, and understanding of pathogenic *Listeria* traits that enable it to fully thrive and protect itself from harsh conditions and environments it encounters, including those in the food chain and within the host itself [43]. This indicates that pathogenic *Listeria* strains are diverse in terms of virulence, host, and ecology. Epidemiological surveillance is imperative, as it mitigates the burden of foodborne illnesses, helps identify emerging contamination sources, and improves prevention strategies. However, there remains a lack of epidemiological and microbiological surveillance of *Listeria* in the food and agricultural industries in South Africa.

## 2. Research Methodology

This systematic review was conducted in compliance with and using the guidelines specified by the preferred reporting items for systematic reviews and meta-analysis [PRISMA] [44] and was not registered (Figure 1).

### 2.1. Eligibility Criteria

#### 2.1.1. Inclusion Criteria

Screening of articles was carried out to specifically identify papers that adhere to the study title and abstract and pertain to *L. monocytogenes* and *L*. *ivanovii* on leafy greens from farms and markets, and to exclude ambiguous studies regarding other *Listeria* species. The journal papers were only deemed relevant and eligible if (a) the source type and document type was a journal article published in English, (b) *Listeria* species such as *L. monocytogenes* and *L. ivanovii* that are identified to the relevant species level by either PCR or genome methods, (c) type of virulence identified including its source and risk parameter, (d) enumeration location which also includes abiotic and biotic parameters leading to proliferation, (e) agronomic parameters and cold chain parameters leading to succession of *L. monocytogenes* and *L. ivanovii* on leafy greens from farms to markets to fork (f) various biocides/disinfectants for mitigation or reduction, tolerance of *L. monocytogenes* and *L. ivanovii*, including strategies utilised within *Listeria* species for protection and survival, and (g) both *Listeria* tolerance and ability to cause foodborne disease.

#### 2.1.2. Exclusion Criteria

Other journal papers that included *Listeria* species other than those mentioned were excluded because they contained ambiguous data and lacked relevant outcomes. Any other fresh vegetables that do not fall under leafy greens were excluded as the study is focused on leafy greens only.

### 2.2. Search Strategies

This review article was developed by examining relevant English-language journal articles on the prevalence, identification, and characterisation of *L. monocytogenes* and *L. ivanovii* on leafy green vegetables at small-scale farms, including their virulence, adaptation, and proliferation, leading to succession at market establishments. Therefore, the authors conducted a review using various databases such as PubMed, Taylor and Francis, Oxford, and Google Scholar with the following keywords:

Pathogenic *Listeria* prevalence on leafy vegetables at farms and markets; strategies and communication within specific *Listeria* species; virulence and adaptations of *Listeria* species; parameters leading to succession in leafy greens; and biocides/disinfectants for reduction and mitigation, including infectious dose.

### 2.3. Selection Process

The authors selected research articles published from 1992 to 2025 that documented the virulence of *L monocytogenes* and *L. ivanovii* in leafy greens from farms and markets. The three-decade dataset will enable surveillance and highlight trend studies on the distribution of leafy greens from primary production through the food supply chain, their associated *L. monocytogenes* and *L. ivanovii* pathogens, and whether a new niche is stable, expanding, or shrinking. This will highlight trends observed across regions and commodity types. These provide clear surveillance of genes detected, whether there are new adaptations or mechanisms adopted by these pathogens over the years, up to recent times. The authors also point out the aetiology and epidemiology of these pathogens and their adaptation differences, including the serotypes associated with leafy greens and the food processing environment, such as farms and markets, across various geographic regions. All reviewed material was analysed and discussed in light of the primary objective, with the conclusion leading to an understanding of small-scale farm contamination and potential risk parameters associated with *Listeria* prevalence, virulence, and succession in markets, implicating leafy green vegetables. Keywords utilised when searching the databases included “*Listeria* infections dose”, “*Listeria* survival or survival strategies”, “*Listeria* virulence”, “disinfectants or biocides” and “*Listeria* adaptations” on leafy greens.

### 2.4. Data Collection Process

Eligible data from various studies included the author’s initials and surnames, publication date, period the study was conducted, diagnostic method that was utilised to identify *L. monocytogenes* and *L. ivanovii* from all other species, prevalence and agronomic parameters leading to succession, number of sampled collected if mentioned as well as virulence factor, including their internalins, proteins and virulence regulator that regulates the pathogenicity of *L. monocytogenes* and *L. ivanovii*, and including their similarities compared to other *Listeria* species. In cases where data was not fully discussed, this was noted and made clear due to a lack of available data on the topic. Every step of the systematic review, including quality assessment, eligibility and inclusion criteria, and data extraction, was carried out by on reviewer. In cases where data were not provided in the article but were available as supplementary material, the supplementary material was downloaded and considered during the screening process. No further attempts were made to contact the authors of journal articles for additional materials or unpublished data.

### 2.5. Data Items

A total of 609 studies were initially identified from databases such as PubMed (*n* = 27), Taylor and Francis (*n* = 10), Oxford (*n* = 10), and Google Scholar (*n* = 557). After excluding 10 duplicates and 58 records deemed irrelevant to the scoping review objective, 529 records were screened. During the initial screening of titles, abstracts, and full texts, 396 studies were deemed eligible for a more in-depth evaluation. Of the remaining 405 reports sought for retrieval, 136 studies were removed, leaving 227 for inclusion in the review. The 405 studies sought for retrieval were relevant, except that some studies report or highlight only one variable regarding survival or virulence of *Listeria*, or one variable of strategy, even though they are from various countries. The studies were not removed for irrelevance but rather because they reported on a single variable related to the review’s objective, rendering them redundant. For example, exploring stress as a strategy parameter for one commodity rather than expressing more variables in one study.

## 3. Results and Discussion

### 3.1. Listeria monocytogenes and Listeria ivanovii Virulence Determinants

In 1999 and 2010, studies reported that *L. ivanovii* was exclusively associated with ruminants, but it was later shown that *L. ivanovii* infections occurred in humans after ingestion of contaminated foodstuffs. It was also concluded that a wide variety of foodstuffs are now a source of this pathogen and, as with *L. monocytogenes*, *L. ivanovii* can persist in food production establishments [14]. *L. monocytogenes* has been characterised by significant diversity and grouped into major lineages, including lineage-serotypes and clonal complexes (CCs) [21]. The *L. monocytogenes* serotypes are based on somatic (O) and flagellar (H) antigens. The serotypes are considered highly invasive and malleable, able to withstand various environmental conditions that contribute to virulence. Designation is associated with virulence potential, while pathogenicity islands (LIPIs) are specific DNA regions that harbour virulence genes, which are crucial for the bacterium’s ability to cause disease. The O antigen is mainly determined by the sugar substituents of teichoic acid, which are bonded to the peptidoglycan of the cell wall, influencing pathogenicity and adaptability [45]. Published articles from various parts of the world have reported that serotypes 1/2a, 1/2b, and 4b are predominant in fresh produce [46,47,48,49]. A study from New York (United States) reported four major lineages and 14 serotype-related lineages with more than 170 clonal complexes [21]. Few studies from Yangzhou (China), Germany, and Paris (France) reported more diverse evolutionary lineage I to include serotype 1/2b, 3b, 4b, 4d, 4e, and 7, lineage II comprises serotypes 1/2a, 1/2c, 3a, 3c, and 4h, lineage III includes serotypes 4a, atypical 4b and 4c, while lineage IV includes serotypes 4a and 4c with somatic and flagella antigens [28,50,51].

For the strains to cause human listeriosis, certain genes must be present in these species and expressed under conducive conditions, such as pathogenicity islands LIPI-1, LIPI-3, and LIPI-4, which encode key virulence factors, and the invasion locus inlAB, which contributes to invasion and protection of *Listeria* [52]. A study in Germany highlighted that LIPI-2 was considered LIPI only for *L. ivanovii* and was mostly present in feed locks, including bovines (*Bos Taurus*), and rarely caused infection in humans [1]. LIPI-3 was mostly isolated and detected from lineage I, which consists of clonal complexes such as CC_1_, CC_2_, CC_3_, and CC_228_ [52]. The pathogenicity of *L. ivanovii* mainly depends on a metalloprotease (*mpl*), pathogenicity island 1 (LIPI-1) and LIPI-2. LIPI-1, which also exists in *L. monocytogenes,* harbours genes for a central virulence regulator (*prfA*), two phospholipases (*plcA* and *plcB*), an actin polymerise surface protein (*actA*), a pore-forming toxin (*hly*), and LIPI-2 is specific for *L. ivanovii* and includes genes coding for a sphingomyelinase C (*smcL*) and several internalins [53]. *Listeria ivanovii* was previously termed *L. monocytogenes* serovar 5 because of some imperative similarities in the pathogenesis between the two species, *L. monocytogenes* and *L. ivanovii*, as they both invade mammalian cells in tissue culture and spread from cell to cell, except that *L. ivanovii* lacks the *L. monocytogenes* cytotoxic factor. It is suggested that this could account for its lower virulence [54]. Regardless of the facts provided regarding *L. monocytogenes*, regulatory bodies consider all strains to be at the same level regarding pathogenicity [12]. Not enough evidence is available to support *L. ivanovii* in line with the objective of the study regarding leafy greens, except for a few studies reporting its emergence in leafy greens.

In Paris, France, LIPI-1, found in *L. monocytogenes*, was 9 kb, located between the genes *prs* and *orfX*. It consisted of six genes, including *prfA*, *plcA*, *hly*, *mpl*, *actA*, and *plcB* [55]. In North Carolina, researchers reported that the *prfA*-virulence gene cluster (*p*VGC) is the main pathogenicity island in *L. monocytogenes*, comprising *prfA*, *plcA*, *hly*, *mpl*, *actA*, and *plcB* genes, which include the internalins (*inlA*, *inlB*, *inlF*, *inlJ*) playing key roles in invasion and adhesion [56]. The virulence genes of *L. monocytogenes* from three locations (Australia, Greece, and Ireland) were reported to follow distinct pathogenicity evolutionary pathways, which are influenced by various factors, including strain type and serotype, leading to dominance [22]. All *L. monocytogenes* serotypes are considered detrimental to humans; certain serotypes, such as 1/2a, 1/2b, and 4b, account for 95% of all described human listeriosis cases [45], while serotype 4b is considered the most pathogenic [49]. Studies from Spain, China, Italy, and the Czech Republic reported that serotypes 1/2a, 1/2b, and 4b are predominant in fresh produce [46,47,49,57]. A study from South Africa using multiplex PCR reported that serotype 1/2b was the most prevalent in fresh produce [40]. *L. monocytogenes* isolates were assessed from the Western Cape province (South Africa) and multiple antibiotic patterns contradicted certain global resistance patterns [58]. The above study reported that all isolates tested were susceptible to ampicillin, while many were resistant to chloramphenicol, erythromycin, and tetracycline. The patterns of resistance observed in this study differ from those observed elsewhere. Moreover, a study determined that laboratory-based surveillance should be enhanced by genomic surveillance techniques like whole-genome sequencing that improve the efficiency of outbreak detection and epidemiological identification of implicated *L. monocytogenes* RTE products and food safety regulations and standards, including limits in high-risk RTE products, will be necessary in South Africa to provide a layer of consumer protection [59]. The *inIA* gene was used for PCR amplification in lettuce and spinach; the results showed higher levels of *L. monocytogenes* at 87 CFU/g and 71 CFU/g, respectively [60]. Both commodities, spinach and lettuce isolates, showed significant resistance to colistin (56.2% and 53.3%, respectively) and to ampicillin (68.8% and 53.3%, respectively). *L. monocytogenes* strains isolated from conventionally and organically grown fresh vegetable markets in Florida and Washington (United States) were characterised and various serotypes isolated included 1/2b, 1/2c, 3b, 3c, 4a, or 4c, all of which were found in conventional fresh produce and were resistant to sulfonamides. At the same time, organic produce was not resistant to either sulfonamides or ciprofloxacin [61]. Furthermore, the *inlA* gene is used as a marker for detecting *L. monocytogenes* in water, including in fresh produce [62]. A study investigating a molecular marker for evaluating the pathogenic potential of foodborne *L. monocytogenes* reported that all 110 strains belonged to serovar 4b, the most frequently implicated serovar in human listeriosis, and that all strains expressed full-length internalin [63]. This study provides a molecular explanation for the predominance of serovar 4b among clinical strains and supports the utility of studying internalin expression as a marker of virulence in humans [63]. The prevalence of sanitiser-resistant *L. monocytogenes* was assessed in South African food processing environments. The chosen isolate carried a lineage II (bcrABC and emrC) resistance gene, conferring resistance to benzalkonium chloride (BAC), which was detected after 15 h of initial treatment. However, no resistance was reported when the isolates were treated with paracetic acid (PAA) and the quaternary ammonium compound (QAC) sanitiser from Byotrol [64]. Collaborating authors from various countries, including Denmark, Spain, Germany, Austria, Norway, Switzerland, the United Kingdom, and Italy, reported that bcrABC, emrE, emrC, and qacH were isolated from the food processing environment, with bcrABC being the most frequently detected QAC gene in the United States. In addition, *qacH* was dominant in Europe and tolerant to QACs, while all isolates showed similar sensitivity to PAA [65]. Table 1 below depicts *L. monocytogenes* virulence genes detected from leafy green vegetables. 

Given frequent cold chain challenges, the potential for *L. monocytogenes* and *L. ivanovii* to grow in contaminated leafy vegetables can further amplify consumer exposure risks. The majority of these studies focus on virulence genes and virulent strains that make *L. monocytogenes* and *L. ivanovii* pathogenic. Most studies highlight inlA and inlB as internalins, while few include inlC as essential, assisting the pathogen in invading the host, even though most studies reported only on inlA as the dominant internalin in most cases. Studies also highlight that the most common *Listeria* serotypes 1/2a, 1/2b, 1/2c, and 4b are considered common and found in leafy greens, while the serotype 4b variant, which is termed 4bv-1, is responsible for most human listeriosis cases.

Pathogen virulence genes found in different types of leafy greens can differ due to genetic variation among strains and to environmental conditions specific to each leafy green, leading to distinct gene expression patterns. In essence, the specific set of virulence genes in the *Listeria* strain is a result of the pathogen’s evolutionary history together with its genetic makeup, while on the other hand, the distinctive and conducive leafy green environment may determine which of these genes are turned on to aid survival by utilising various mechanisms for adaptation and eliminating competition, leading to potential infection. 

### 3.2. From Farm-to-Fork Continuum: Potential Sources of Contamination for Listeria

Minimally processed green leafy vegetables are generally labelled as any leafy vegetable that has been subjected to different processing stages, including trimming, cutting, washing or disinfection, rinsing, and storage [51]. Leafy green commodities, phylloplane topography, and natural apertures, including agronomic activities, influence the diversity of the microbial community [74]. Leafy green vegetables’ water level generally ranges from 0.970 to 0.996 and high-water content supports microbial growth and spoilage, since microbes require moist environments to thrive and multiply [75,76]. The waxy cuticles, including the internal leaf tissue of the phyllosphere, and other essential polysaccharides serve as protective factors against pathogenic bacteria by keeping disinfectants and other environmental stresses at bay [77,78]. Ref. [79] described spinach (*Spinacia oleracea*) to have a large amount of β-carotene in its phyllosphere and an extensive community of bacteria, including extracellular polysaccharides produced as a principal component of the formation of biofilm. Contamination can take place during harvesting of fresh leafy vegetables, during preparation, in treated wastewater, in livestock manure, during washing, in cold chains, in retail, and even in the last stage, the consumer’s kitchen [80]. The transfer of pathogens from one host to another is influenced by various traits of both the pathogen and the environment, including extrinsic factors.

South Africa is a water-scarce region, and its agricultural sector is highly dependent on municipal water or rainfall for irrigation. A study in the Eastern Cape province (South Africa) between August 2007 and July 2008 investigated antibiotic susceptibilities and the distribution of *Listeria* pathogens in the final effluents of a rural wastewater treatment facility, including the water received from the watershed [81]. The study reported that free-living *Listeria* were more prevalent (96%) than plankton-associated *Listeria* species (58–67%). Of 56 samples, 11 (19.6%) were confirmed as *L. ivanovii*, and 1 (1.8%) as *L. monocytogenes* [81]. Review studies focused on the irrigation of crops with municipal wastewater [82], including health implications of wastewater re-use in vegetable irrigation [83]. Ten leafy greens and produce farms in New York State (United States), reported *L. monocytogenes*, *Listeria* spp., *Salmonella*, and STEC to be detected in 16, 44, 4, and 5% of terrestrial samples, 30, 58, 12, and 3% of water samples, and 45, 45, 27, and 9% of faecal samples, respectively, suggesting that intervention at the irrigation level may reduce the risk of produce contamination [84]. Weekly irrigation water samples from four vegetable farms in the United States also showed that *Staphylococcus*, *enterica*, and *L. monocytogenes* persisted in irrigation water and have been linked to produce contamination events [85]. Another study in the Eastern Cape province (South Africa), in the Bartman district municipality, highlighted multidrug-resistant *Listeria* species, which were abundant in isolated samples from the river and irrigation water [38]. Of the confirmed species, 41 isolates (59%) were classified as *L. monocytogenes,* and all *Listeria* spp. exhibited phenotypic resistance against ampicillin, penicillin, and trimethoprim-sulphamethoxazole, and further screening revealed antibiotic resistance genes in the following proportions: *sulI* (71%), *blaTEM* (66%), *tetA* (63%), and *blaCIT* (33%), confirming the occurrence of antibiotic resistance genes among *Listeria* inhabiting surface waters. Contrary to reports on the reuse of municipal wastewater for irrigation, wastewater is considered by some an alternative water source, as the agricultural sector is the largest water user worldwide [86]. The study applied a choice modelling approach to identify the elements defining frameworks and to quantify their relative importance amongst farmers in the agricultural hinterland of Cape Town. The findings suggest that water reuse is acceptable to farmers. This is a threat because it poses a significant concern and human risk. If conducive conditions exist, *Listeria* spp. can grow, colonise, and form a biofilm due to its complex traits, which allow it to protect itself from various conditions. If the food is ingested, the toxin is released, and then food poisoning can occur. Table 2 below depicts *L. monocytogenes* and *L. ivanovii*, including their infective doses from various food sources ingested.

Standards and limits for *L. monocytogenes* in foods were set by the regulations of some African countries [59]. The inability of the government of South Africa to effectively regulate the food sector is a contributing factor to increased food safety risks [41]. Europe’s fresh produce guidelines stipulate that *L. monocytogenes* should not be present (<1 CFU 25 g^−1^), especially in ready-to-eat products that support its growth [89]. Other countries across the world, including Austria, Belgium, Denmark, Finland, France, Germany, Greece, Ireland, Italy, the Netherlands, Norway, Portugal, Spain, Sweden, Switzerland, the United Kingdom, and others, have implemented and adopted less stringent guidelines, allowing a small amount of contamination [30,90]. Several large outbreaks of a febrile gastroenteritis syndrome have further highlighted the importance of considering *L. monocytogenes* to be a foodborne pathogen. In these outbreaks, with an average incubation period of approximately 24 h, attack rates up to 72% were much higher than those reported for outbreaks of invasive listeriosis [30]. It has been demonstrated by [91] that *L. monocytogenes* can survive for at least eight weeks in the environment. Other studies reported that *L. monocytogenes* persists for months in food processing facilities due to its ability to survive under stressful conditions, including heat, high salt content, desiccation, and refrigeration [92,93]. For example, the study reported that *L. monocytogenes* and *L. ivanovii* could survive in leafy vegetables from farm to market, and that the prevalence and succession of pathogens were studied [66]. Scientific reports are important because they help fill gaps and enhance epidemiological data on food safety and consumer protection.

The roles of virulence and surface proteins such as *SigB*, *PrfA*, *ActA*, *InlA*, *InlB*, *InlC*, and *InlH*, which are from the LPXTG family, regarding *L. monocytogenes* pathogenesis, suggest that it utilises a specific host–parasite interaction that is mediated by a specific interaction between its surface proteins and host cell receptors [94,95]. Additionally, *L. monocytogenes* was reported on non-food-contact surfaces and objects, including floors, drains, sinks, and walk-in cooler shelves, in market facilities [96]. *L. monocytogenes* proliferation was reported due to temperature fluctuations in cooler environments during the distribution stage and commercialisation of food products in California, North Carolina, Arizona, Texas, and Ohio (United States) [97]. In the south-eastern United States, eleven fresh produce packinghouses were surveyed, and 2.64% of samples were positive for *Listeria* species and 3.15% for *L. monocytogenes*, with positive samples found in drains, cold storage rooms, and outside packing/handling areas [98]. In China, 23 (5.49%) of 419 vegetable samples were positive for *L. monocytogenes*, where lettuce was highest (7.78%), followed by coriander (*Coriandrum sativum*—4.49%), where the most common phylogenetic group (1/2a-3a) and II.2 (1/2b-3b-7) strains were ST_87_ (36.7%) and ST_8_ (26.7%). A virulence analysis showed that all 30 isolates harboured eight classical virulence genes, 10.0% isolates harboured the *llsX* gene (ST3 and ST_1_ strains), and 36.7% carried the *ptsA* gene and belonged to ST_87_. Approximately 83.3% isolates carried full-length *inlA*, whereas five isolates had premature stop codons in *inlA*, three of which belonged to two ST_8_ and ST_9_ [8]. The test results for 136 samples (15.1%) of vegetable salads, cabbage, and lettuce, including spinach, parsley (*Petroselinum crispum*), coriander, and dill (*Anethum graveolens*), collected from the Mazandaran and Golestan provinces in northern Iran, were positive for *Listeria* spp., with green vegetables accounting for 23%. The dominant serogroups were 1/2a and 4b. Furthermore, all isolates harboured four virulence genes, including *hlyA*, *plc*, *iap*, and *actA* [69]. Another study in São Paulo, Brazil, assessed the survival and growth behaviour of *L. monocytogenes* over storage time, which varied with temperature, across 14 different types of ready-to-eat vegetable salads. The lettuce, purple cabbage, and white cabbage, along with other produce, were stored at 4, 8, 12, and 16 °C for up to 144 h. The white cabbage supported growth only at 16 °C, whereas all salad products containing lettuce as a constituent vegetable supported the growth of *L. monocytogenes* at 12 and 16 °C [99]. A study on salad leaf fluid content that included iceberg lettuce, spinach, rocket, and mixed salad leaves (lettuce, spinach, radicchio [*Cichorium intybus*] mixtures) and their ability to support the growth of *L. monocytogenes* in sandwiches [100] reported that the compounds released from damaged salad leaves can act as potent stimulators of *Listeria* growth, biofilm formation, and virulence, which can cause infection, highlighting that the strain EGD is also of the same genetic lineage as the 2019 United Kingdom hospital outbreak isolate.

A study in the Free State province (South Africa), detected *L. monocytogenes* and *L. ivanovii* in spinach isolates from two small-scale farms, and their succession was also detected in two spinach markets, with possible contamination from livestock manure [66]. The farms where these pathogenic *Listeria* species were detected are the major small-scale farms that supply most local markets within the district with leafy greens. *Listeria* spp. can grow at low temperatures, and when organisms enter a stationary phase, the period of cold storage essentially serves as a selective enrichment for this species [101,102]. Thus, refrigeration temperatures will not totally retard microbial spoilage, as these are favourable conditions for some microorganisms, such as *Pseudomonas* spp. and *Listeria* spp., which rapidly grow under such temperatures [103]. Failure to keep leafy green perishable vegetables within the desired temperature range (0–2 °C) due to insufficient refrigeration can stimulate the growth of potential pathogens [104]. The same O_2_-depleted atmospheres at 0.25% utilised to control enzymatic browning in lettuce (*Lactuca sativa*) enhanced *Listeria* growth in fresh-cut iceberg lettuce stored at mild abuse temperatures [105].

*L. ivanovii* has been reported in numerous animal and plant foods, mostly from African countries, such as Ethiopia, Nigeria, and, predominantly, Egypt [106]. The high prevalence was reported to be higher in countries where small ruminant farming is widely practised; however, in countries such as Australia and New Zealand, nothing was reported [14]. This could be a result or an indication of under-reporting of *L. ivanovii* prevalence [14]. *L. monocytogenes* was detected in cattle manure, animal feeds, soil, and growing grass in four urban areas in New York City and Austria, respectively [107,108]. *L. monocytogenes* was isolated in ready-to-eat, post-processed foods in market establishments, where bacterial transfer may be due to inadequate post-processing procedures from the farm [109]. Lastly, the distance travelled and long storage must be considered, as they affect the desired temperature range during this critical step [103]. It is also important that the inside of the cooling systems is regularly maintained and properly designed to transport fresh produce, thereby reducing the potential for microbial contamination and proliferation [110].

### 3.3. Agronomic and Market Parameters Leading to Prevalence of Listeria spp.

Supermarkets have been spreading rapidly across developing countries over the past decade. The rise in supermarkets was most significant in South Africa, Kenya, and Nigeria [111]. Each province in South Africa is unique regarding the agricultural commodities it can produce. Awareness and education about the presence of *L. monocytogenes* at the retail/supermarket level are critical, as this is the last step before the product reaches the consumer. The informal sector of fresh produce supply is now becoming prevalent in urban South Africa [112]. Consumer participation in informal vegetable markets is vital to the urban economy, as they offer easy access to food and because informal traders source their supplies [113]. Many markets obtain their vegetables from small-scale farms due to accessibility and discounts (particularly when purchased in bulk), and because farms also permit pick-your-own. Most consumers (79% of 99.5%) purchase more perishable vegetables from informal traders due to the traders’ convenient locations [114]. Markets are essential and play a vital role with regard to easy access, but consumers might be risk owing to compromised hygiene status which may lead to an illness. Consumer awareness and education on food safety and preventative measures is important to save a life.

#### 3.3.1. *Listeria* Sources Associated with Leafy Vegetables from Small-Scale Farms

The source of contamination or presence of *Listeria* linked to leafy greens grown on small-scale farms includes biotic and abiotic factors, livestock manure, irrigation water, and poor hygiene. Minimal processing operations are known to trigger the onset of many physiological changes, thereby reducing product quality [115]. Many sources introduce enteric pathogens into soil, including wastewater, raw manure, compost, human biosolids, wild and domestic animal intrusion, and other anthropogenic activities [116]. In Nigeria, bacterial organisms such as *L. monocytogenes*, *Escherichia coli* O157:H7, and Salmonella enterica are the most common pathogens associated with contamination of food crops grown with organic fertilisers [117,118,119,120]. In France, the presence of *Listeria* spp. and *L. monocytogenes* was established in various media, such as sewage sludge, confirming the presence of *L. monocytogenes* at all stages of treatment, with 73% from dewatered sludges, 80% from stored sludges, and 95% from activated sludges [121]. *Listeria* spp. are saprophytes that are present in soil across various agricultural landscapes. Freshly harvested produce in close proximity to the topsoil is considered a high risk for *Listeria* spp. contamination [108,122]. Several studies reported on human enteric pathogen internalisation via root uptake into leafy greens [123,124], and on plants as a realised niche for *L. monocytogenes* and a risk of human pathogen internalisation in leafy vegetables [125,126]. In the Pacific northwest of the United States, it was reported that seven fresh produce handling and processing environments detected positive for *L. innocua*, *L. ivanovii*, and *L. welshimeri* strains in drains, equipment legs, forklift tyres, entry points and floors, with the represented serotypes being 1/2a, 3a, 4b, 4d, and 4e utilising a modified version of multiplex polymerase reaction for molecular sero-grouping of isolates [127]. Similarly, previous studies found that wild-type *L. ivanovii* samples were more invasive than clinical *L. monocytogenes* [15]. Another study in South Africa reported that of five fresh leafy greens farms analysed, only two farms in Kroonstad and Henneman were reported to have *Listeria* spp. (*L. monocytogenes* and *L. ivanovii*) on spinach [66].

The Kroonstad farm’s *L. ivanovii* contamination may be from livestock manure, as they were also producing livestock and utilising manure as fertiliser for their vegetables. The Henneman farm’s contamination may be due to poor hygiene practices and inadequate sanitation design resulting from the infrastructure. The study also enumerated the prevalence of *L. ivanovii* and *L. monocytogenes* in two retail stores selling spinach (*Spinacia oleracea*) in two towns, Ficksburg and Henneman [66]. In this case of succession, the study confirmed the possibility of contamination emanating from poor sanitary practices during processing and handling at the processing facilities, leading to the succession of bacteria at retail. Livestock are considered potential primary sources and silent carriers of pathogenic *Listeria* species, resulting in pathogen dissemination via faeces into the environment, equipment, farm surfaces, and food processing plants [128,129]. The emergence of *L. monocytogenes* and *L. ivanovii* in South Africa, particularly among small-scale farms growing leafy greens, could pose a challenge in the near future if left unchecked. Table 3 below depicts sources of contamination emanating from various small-scale farms linked to leafy greens.

Generally, the most common and easily discernible sources of contamination in most studies are irrigation water, soil, and fertiliser used on leafy greens at various farms. Most studies report potential sources of *L. monocytogenes* and *L. ivanovii* contamination in leafy greens emanating from pre-harvest and post-harvest sources on the farm, followed by cold chain disruption and poor hygiene at markets. The studies highlight that it is therefore imperative to fully comprehend the pathogen-contamination routes and the various parameters that contribute to contamination risk, prevalence, spread, and the emergence of new niches, as they are the building blocks of an effective control programme.

#### 3.3.2. Shift in Bacterial Composition to a New Niche at Markets

The preservation, storage, and transportation of perishable vegetables are managed through a cold supply chain to slow biological decay and deliver safe, high-quality foods to consumers [103,145]. Histidine and kinases of *L. monocytogenes* are considered two key components of a system responsible for growth and adaptation to cold stress, including low temperatures [146,147,148]. Additionally, the authors reported that the two components mentioned above are involved in bacterial low-temperature responses, including the genes yycGF and lisRK, which are involved in cold stress adaptation. Furthermore, the study highlighted that the protein encoded by these genes was involved in early stages of bacterial survival, while *lisRK* is involved in cold acclimation. Moreover, glycine, betaine, and carnitine produced by bacterial cells accumulate from the environment via the chill-activated transport system. These osmolytes are also found in food, promoting the growth and survival of *L. monocytogenes* at lower temperatures.

Fresh produce is susceptible to pathogen survival and growth due to poor handling, which creates opportunities for contamination, growth, and ingress into plant tissues [149]. Policies and standards must address potential sources of contamination on produce, such as jewellery, hair, and beards, which pose hazards from employees who may also harbour faecal matter and pathogens [150]. Refrigeration temperatures will not totally retard microbial spoilage, as these are favourable conditions for some microorganisms, such as *Pseudomonas* spp. and *Listeria* spp., which grow rapidly under such temperatures [103]. Failure to store perishable food at the desired temperature range due to insufficient refrigeration can promote the growth of potential pathogens [104]. A study in Spain assessed the effects of oxygen-depleted atmospheres on the survival and growth of *L. monocytogenes* on fresh-cut iceberg lettuce stored at mild-abuse commercial temperatures and reported that the very O_2_-depleted atmospheres of 0.25% utilised to inhibit enzymatic browning are well able to enhance *Listeria* growth in fresh-cut iceberg lettuce [105].

Small-scale growers who sell their produce locally to consumers use their own vehicles for most farm purposes. Transportation is considered an imperative post-harvest process that occurs not only between the main stages but also along the food supply chain, where fresh vegetables are distributed to final destinations. The vehicles and containers from these small-scale growers to transport fresh produce could also be potential sources of contamination [151]. Quality loss in fresh produce is a function of both time and temperature, particularly at this stage prior to distribution, when the product might be neglected during loading or off-loading, compromising shelf-life [152]. Lastly, the distance travelled and long storage must be considered, as they affect the desired temperature range during this critical step [103]. It is also important that the inside of the cooling systems is regularly maintained and properly designed to transport fresh produce, mitigating the potential for microbial contamination and proliferation [110]. Education and awareness about *Listeria* prevalence and contamination at the market level are imperative, as this is the final stage before fresh produce reaches consumers.

In food processing establishments, *L. monocytogenes* is exposed to many harsh environmental conditions; thus, the importance of *σB* activity for maximum survival under surfactant stress, both outside and within the host, is evident [153]. *L. ivanovii* pathogenicity depends on two pathogenicity islands, namely LIPI-1 and LIPI-2, of which LIPI-1 exists in *L. monocytogenes,* and LIPI-2 is specifically for *L. ivanovii* and consists of genes coding for sphingomyelinase C (*smcL*), including several other internalins [53]. Additionally, all pathogenicity island 1 and 2 genes are regulated by *prf*, a central virulence gene [153]. Virulence key factors of *L. monocytogenes* depend on *prfA* integrity, which positively and coordinately regulates transcription of several virulence genes. The pattern of *PdPs,* which are *PrfA*-dependent proteins expressed in *L. ivanovii,* regulating virulence genes, was similar but not identical to that of *L. monocytogenes* [154,155]. Screening and characterisation of nonhemolytic mutants obtained by transposon mutagenesis indicated that hemolysin expression is critically needed for *L. monocytogenes* virulence [156]. The *PrfA* gene is a virulence gene also transcriptionally regulated by *σ^B^*, which controls the general stress response in *L. monocytogenes* [156]. Additionally, the *Ilo* hemolysin of *L. ivanovii* and the *Llo* hemolysin in *L. monocytogenes*, CDTX, are cholesterol-dependent pore-forming toxins that are essential for the intracellular cycle of the pathogens, allowing lysis of the vacuole in the infected cell. This is the key step in the proliferation and spread of infection to neighbouring cells. *L. ivanovii* is also capable of lysis of the host cell phagosome and actin polymerisation but is less effective than *L. monocytogenes* in cell-to-cell spread and intracellular multiplication [157]. Harsh conditions that affect the cell reduce membrane lipid fluidity. In response to low temperatures and other conditions, *L. monocytogenes* alters membrane lipids; these lipids are essential for the optimal structural and functional integrity of the membranes, as they prevent a gel-like state that can cause leakage of cytoplasmic contents [6]. The mechanisms of this phenomenon are complex and involve decreased bacterial cell metabolism, changes in cell membrane composition, expression of cold shock proteins, and the uptake of cryoprotective compounds from the environment [158]. Upon cold shock, *L. monocytogenes* dramatically reduces its growth rate and induces enzymes participating in the synthesis of precursors of branched-chain fatty acids, and transporters of glycin-betaine (*gbu*), carnitine (*opuC*), and oligopeptides (*oppA*), which may contribute to maintenance of membrane fluidity and increase in the uptake of compatible solute [159]. Another protein responsible for cold shock and molecular chaperoning is *CspA*, which enables protein synthesis at low temperatures and melts RNA secondary structures. Two *Listeria CspA* proteins, *CpsB* and *CpsD*, are downregulated during this process [159]. Among all the *cps* identified, *CspA* contributes to *L. monocytogenes* resistance to harsh conditions, such as low temperature [160]. Contrary to the resistance of *L. monocytogenes* to harsh conditions [161], UV-C (1.3 kJ m^−2^) on lettuce induced a stress response in the plants that reduced *L. monocytogenes* attachment, survival, and growth at pre-harvest. Further exploration of this technique may enhance the microbial safety of lettuce.

### 3.4. Adaptations and Proliferation of Pathogenic Listeria in Leafy Green Vegetables

Survival of *L. monocytogenes* across ecosystems is key to its transmission to different commodities; in essence, *L. monocytogenes* can persist in a food processing facility for months and re-contaminate other products [162]. *Listeria monocytogenes* is known to interact with the roots of leafy green vegetables, colonise and internalise into mature plants, including seedlings, and has been isolated from contaminated fresh produce [163]. Precise identification of a potential contamination source for leafy green vegetables is often difficult because contamination can occur at any point along the farm-to-fork continuum [164]. Another study in China reported that 5.49% of the 419 vegetable samples tested contained *Listeria*, and that virulence genes such as *prfA*, *mpl*, *plcA*, *inlB*, *plcA*, *hlyA*, *iap*, and *actA* were carried by all strains of *L. monocytogenes* [8]. Moreover, 10.0% carried the *llsX* gene, which belongs to ST_3_ and ST_1_ strains; 36.7% carried the *ptsA* gene, which belongs to ST_87_; and 83.3% of the isolates were full-length for *inlA*, which belongs to ST_9_ and ST_8_, clearly pointing out that most isolates were capable of penetrating and invading host cells. In Shanghai (China), four *L. monocytogenes* serotypes, such as ST_5_, ST_121_, ST_120_, and ST_2,_ considered virulent, have been identified in two ready-to-eat foodstuffs from 2019 to 2020, with ST_5_ being the predominant in one ready-to-eat food processing plant [165]. In South Africa, ST_121_ and ST_321_ are considered hypovirulent, while ST_1_, ST_2_, and ST_204_ are hypervirulent strains that could pose a major public health risk in food processing environments and meat products [166]. Prevalence, phenotypic and genotypic characteristics of *L. monocytogenes* was isolated from ready-to-eat vegetable market in São Paulo (Brazil), lettuce (*n* = 152), collard green (*Brassica oleracea* L.) (*n* = 24), arugula (*Eruca sativa* L.) (*n* = 19), mix for yakisoba, containing cauliflower, carrot, broccoli, cabbage, and chard (*n* = 18), watercress (*Nasturtium officinale*) (*n* = 18), chicory (*Cichorium intybus* L.) (*n* = 16), escarole (*Cichorium endivia*) (*n* = 13), cabbage (*n* = 11), spinach (*Tetragonia tetragonioides*) (*n* = 11), and isolates were characterised for their serotypes, ribotypes, positivity for virulence genes *inlA*, *inlC* and *inlJ*, resistance to chlorine, growth rate variability and capability to form biofilm. *L. monocytogenes* was detected in 3.1% of the samples, and only five samples presented countable levels, with counts between 1.0 × 10^1^ and 2.6 × 10^2^ CFU/g belonging to serotypes 1/2b or 4b, and most samples were positive for genes *inlC* and *inlJ* [48].

Biofilms mainly comprise a legion of bacterial cells embedded in a self-produced extracellular matrix consisting of compounds such as extracellular DNA (eDNA), carbohydrates, and proteins [167]. Bacterial formation provides extensive protection against detrimental environmental parameters, including disinfectants and desiccation, and provides nutrients. The stress-adaptive response of *L. monocytogenes* is important for survival and is linked to its pathogenesis, which is regulated by sigma factor B (*σ*^B^) and virulence by positive factor A (*PrfA*) [155]. For example, adaptive stress tolerance can respond to acid and various stresses in the gastrointestinal tract (GTI), directly aiding its virulence. Various studies have also shown that the biofilm matrix of monocytogenes can diffuse antimicrobials, resulting in lower antimicrobial concentrations due to its complex matrix, and that it can colonise mono- or multi-species *Pseudomonas* biofilms [168,169,170]. Hydroponic and soil-grown lettuce leaf extracts were able to enhance the survival, growth, and biofilm formation of *L. monocytogenes* on stainless steel coupons representing surfaces in lettuce processing plants [161]. The study reported that *L. monocytogenes* could colonise and form biofilms on lettuce regardless of the growth system used.

Efflux pumps, quorum sensing for communication and coordination within bacterial populations, and the ability to enhance access to nutrient-rich niches also play a role in biofilm formation; they can expel a wide spectrum of antibiotics, heavy metals, metabolites, toxins, and biocides from cells [171]. Resistance to food-processing sanitisers and heavy metals in *L. monocytogenes* from British Columbia (Canada) was assessed, including profiling the antibiogram of clinically relevant *L. monocytogenes* from British Columbia and Alberta (Canada) [172]. The study reported that 17 isolates were resistant to quaternary ammonium compounds (QUATs; 10 µg/mL), with all positive for one known resistance determinant (*bcrABC*, *n* = 16; *emrE*, *n* = 1). Resistance to cadmium (Cd) and arsenic was found in 89% and 24% of the isolates, respectively. *Listeria monocytogenes* is a naturally competent organism that employs conserved strategies to regulate its competence components. For example, *Lde*, *EmrE*, and the *MdrL* efflux pumps may confer disinfectant resistance alongside other mechanisms [173], while *MdrM* and *MdrT* efflux pumps facilitate persistence and replication of *L. monocytogenes* within the gastrointestinal tract, counteracting the bactericidal effects of mammalian bile [174]. *Listeria monocytogenes* contains *Csps*, structurally related small proteins of 65–70 amino acids that bind nucleic acids and regulate and standardise the expression of various genes, including those involved in motility, stress resistance, cellular aggregation, and virulence [175]. These small, related proteins stabilise the nucleic acid conformation and simultaneously act as key molecular chaperones, facilitating transcription, translation, and replication during temperature fluctuations, even at low temperatures [6]. Studies in Zurich (Switzerland) and Mumbai (India), reported that among the many *Csp* proteins identified in *L. monocytogenes*, *CspA* is responsible for the bacteria’s resistance to low temperatures [160,176,177].

Reprogramming is an essential key to adapting to new stresses across various niches of the transcriptional landscape, aligning gene expression with the physiological needs of the cell, and this activity is accomplished by a panoply of proteins, including ribonucleic acid transcriptional regulators [43]. At the pinnacle of the transcriptional regulation hierarchy lie the sigma factors. These sigma factors determine which genes are transcribed by directing the transcriptional machinery to the appropriate promoter sequences. A large number of *L. monocytogenes* strains harbour about five sigma factors, including the principal housekeeping sigma factor *σA* and four alternative sigma factors, *σB*, *σC*, *σH*, and *σL* [178,179]. The *σB* is considered to be the factor that controls the bacterial conditions, such as stress response in *L. monocytogenes*, and of the four alternative sigma factors, it consists of the largest regulon with close to or almost 300 genes, which is approximately 10% of the genome under the positive control of this sigma factor [180]. Genes characterised under *σB* control are known to contribute to a variety of stress resistance mechanisms, including osmoregulation and bile acid tolerance [181]. For example, in *L. monocytogenes*, *σ*^B^ is the major transcriptional regulator of stress response genes, plays a vital role in resistance to detergent stress at lethal levels, and is an essential gene in the activation of biofilms with increased resistance to disinfectants [182]. *σ*^B^ also plays a vitally important role in soil survival by regulating the stress response after *L. monocytogenes* enters the soil, allowing the bacteria to stop multiplying due to nutrient limitation [183]. This phase is similar to entry into the stationary phase, causing *PrfA* to be downregulated and subsequently deactivating key virulence factors, while genes involved in mobility, chemotaxis, and carbohydrate transport are upregulated [183]. 

For example, a study in Ireland demonstrated the endogenous soil microbiota’s suppressive effect on *L. monocytogenes* survival in soil using a pathogen death rate model and showed that the suppressive effect on *L. monocytogenes* survival by the native soil microbiota increases with an increasingly diverse population [184]. Another study in South Africa reported that high counts of *L. monocytogenes* in irrigation water (mean: 11.96 × 10^2^ CFU/100 mL; range: 0.00 to 56.67 × 10^2^ CFU/100 mL) and agricultural soil samples (mean: 19.64 × 10^2^ CFU/g; range: 1.33 × 10^2^ to 62.33 × 10^2^ CFU/g) were documented. Consequently, a high annual infection risk of 5.50 × 10^−2^ (0.00 to 48.30 × 10^−2^), 54.50 × 10^−2^ (9.10 × 10^−3^ to 1.00) and 70.50 × 10^−2^ (3.60 × 10^−2^ to 1.00) was observed for adults exposed to contaminated irrigation water, adults exposed to contaminated agricultural soil and children exposed to agricultural soil, respectively [39]. Another study found that virulence factors such as the cytolysin (listeriolysin O), *ActA* (an actin polymerisation protein), and phospholipases are important for the pathogen’s intracellular survival and the spread of infection [155].

### 3.5. Tolerance of Listeria to Disinfectants Utilised at Leafy Green Minimal Processing

One of the most important preventive measures in terms of tools is the effectiveness in guaranteeing food safety [185]. Disinfectants disrupt bacterial chemical bonds and damage bacterial cellular compounds, leading to a loss of bacterial components and lysis. The prevalence, survival and persistence of *Listeria* strains in foodstuff processing facilities or environments occur mainly because of the bacteria’s efflux pump and biofilm formation [165], and the deletion of *agrA* is considered one method to impair the early biofilm formation of *L. monocytogenes*, whereas the deletion of *agrD* is reported to reduce biofilm formation [186]. Several studies identified the persistence of *Listeria* and *L. ivanovii* from packinghouses, indoor production, and processing, including a fresh produce vegetable farm, where 56.41% *L. monocytogenes* and other *Listeria* species, including *L. ivanovii* 1/2a, 1/2b, 1/2c, 3a, 4b, 4c, 4d, *IVb-v1* serovars identified [187]. *L. monocytogenes* exhibited greater growth during simulated retail storage along the time and temperature profiles where the mean temperature was higher compared to that of transport and retail display, such as −0.3–7.7 °C, 0.6–15.4 °C, and 1.1–9.7 °C during transport, retail storage, and retail display, respectively [97]. This suggests that exposure of *L*. *monocytogenes* to environmental stresses induces a stress response, which is mediated by the alternative sigma factor *σB*, and regulates several stress, virulence, and transporter-associated genes, such as lmo2230, ltrC, ctc, inlA-E and the opuC operon, including related proteins.

Conventional disinfection techniques frequently utilised include oxidative disinfectants, such as chlorine compounds and peroxyacetic acid, as well as non-oxidising disinfectants, such as quaternary ammonium compounds. Quaternary ammonium compounds are, without a doubt, the most commonly utilised disinfectant agents in the food establishment for safety and are efficient against bacteria, algae, fungi, spores, viruses, and mycobacteria, even at low doses [188]. Another study in Daejeon (Republic of Korea) reported that cetylpyridinium chloride at 80 mg/L for 3 min reduced *L. monocytogenes* by up to 4.54 log CFU/g in the spinach phyllosphere [189]. Quaternary ammonium compounds are considered active against bacterial membranes, disrupting the phospholipid bilayer and, concomitantly, causing leakage of cellular contents, leading to bacterial death. In New York (United States), two *L. ivanovii* strains detected from post-harvest sources in fresh produce processing were able to adapt to levels of the disinfectant agent benzalkonium chloride three-fold higher than those of the non-adapted wild types [190]. Several studies, including those in Ontario (Canada) and Warsaw (Poland), have highlighted the resistance of benzalkonium chloride in *Listeria* [191,192]. The resistance is caused by efflux pump genes located in mobile genetic elements, such as *qacA Ide*, *qacH*, *mdrl*, *and emrE*, as well as the *bcrABC* cassette from different ST and CC of *L. monocytogenes* [192]. *L. monocytogenes* biofilm formation increases tolerance to quaternary ammonium compounds by increasing membrane hydrophobicity, promoting further adherence to objects or surfaces [193,194,195].

In Brazil, the *L. monocytogenes* resistance genes *mdrL*, *lde*, *emrE*, *bcrABC*, *radC*, *qacA*, *qacC*/*D*, *qacH*, *qacE*Δ*1*, *cadA1*, *cadA2*, *cadA3*, *cadA4*, and *cadC* isolated from food and food processing identified *MdrL* and *Lde* in 12 (24%) and 33 (66%) isolates and were found to be resistant to BC [196]. The analysis of resistance to BC in the presence of reserpine, an efflux pump inhibitor, showed that efflux pumps did not influence the resistance, while another study highlighted that *L. monocytogenes* resistance genes *MdrL* and *Lde* are members of the major facilitator family and reported that, after addition of reserpine, the MIC for the different strains decreased, indicating that efflux pumps play a role in the adaptation of *L. monocytogenes* to BC [190,197]. Quaternary ammonium compounds and peroxyacetic acid have been widely used to curb contamination, and chlorine compounds have been reported to reduce microbial load in food [198]. In the United Kingdom, chlorine was shown to be ineffective at killing viable but nonculturable *Listeria*, suggesting that *Listeria* can avoid detection by industrial disinfectants while retaining its ability to cause illness or disease. Mobile genetic elements can be exchanged between bacteria and their surroundings, increasing the resistance of *L. monocytogenes* strains, leading to the creation of novel resistance strains and phenotypes [126]. Infiltration of bacteria into produce may be due to a pressure differential, as immersion of warm vegetables in cold water forces cold water directly into the surface apertures of the produce, leading to infiltration [199]. Colonisation of bacteria is a dynamic process in which various factors promote contact, attachment, cell–cell interactions, defence against biocidal wash treatments, and protection against stresses, culminating in the final stage, when bacteria disperse into single cells or new colonisation occurs [200]. Other studies from Gangwon (Republic of Korea) and Ghana recommended a combination of disinfectants to reduce the microbial load of *L. monocytogenes* and enhance fruit and vegetable food safety [201,202]. Another study demonstrated ultrasound of 28 to 68 kHz with NaOCl solutions from 20 to 100 mg/L for 10 min to reduce an average of 0.7 log CFU/g of *L. innocua* population in fresh-cut Chinese leafy cabbage (*Brassica oleracea*), concluding that both treatments displayed synergistic impact on reduction of 85% of *Listeria* on cabbage [202]. A 200 mg/L sodium hypochlorite and a solution of peracetic and percitric acids showed the best performance in reducing 2 and 1.5 log CFU/g in *L. monocytogenes* in raw leafy green vegetables in Padova (Italy) [203]. A study in Poto (Portugal), reported that fresh lettuce (*Lactuca sativa*), rocket salad (*Eruca sativa*), parsley and spinach leafy green vegetables, washed with pediocin, presented significantly (*p* < 0.01) lower microbial load throughout storage, reducing 3.2 and 2.7 log CFU/g, compared to leafy green vegetables washed with water and chlorine, respectively [204]. In most cases of vegetable disinfection to reduce microorganisms for safety, synergistic effects have been shown to improve load reduction. Disinfectants or biocide utilised to reduce microbial load should be at the lethal concentrations recommended [205,206,207].

### 3.6. Strategies and Sensing Coordination for Persistence Within Listeria Species

Since *L. monocytogenes* is regulated by *PrfA*, it also carries a gene cluster of five genes termed stress survival islet 1. These genes contribute to its survival under suboptimal conditions, such as high salt concentrations and pH in food environments [208]. For example, the stressosome (SSI-1) is a complex structural mechanism associated with the alternative sigma factors *σB*, *σC*, *σH*, and σ*L* and is involved in a strong stress response that enables a pathogen to survive and persist in food and withstand passage through the human gastrointestinal tract [6]. In contrast, elevated temperature is associated with increased expression of heat shock proteins (*Hsp*). Furthermore, *Hsp* in *L. monocytogenes* stabilises proteins, prevents improper folding and aggregation, and stimulates repair of denatured proteins. Class I (groE, dnaK, dnaJ, groEL, and groES) is overexpressed during the accumulation of denatured proteins in the cytosol and acts as an intracellular chaperone [6]. Quorum sensing, which is utilised for communication and coordination, influences all types of bacteria. They can be classified into at least three classes: *LuxI/LuxR* systems found in Gram-negative bacteria utilise acyl-homoserine lactones (AHLs) for oligopeptide two-component-type quorum sensing; in Gram-positive bacteria, small AI peptides and *luxS*-encoded AI-2 are utilised [209]. Multidrug efflux pumps are categorised into six families based on topology, structure, and energetics: the resistance-nodulation–cell-division (RND) superfamily, the major facilitator superfamily (MFS), and the proteobacterial antimicrobial compound efflux (PACE) family, the small multidrug resistance (SMR) family, the ATP-binding cassette (ABC) superfamily, and the multidrug and toxic compound extrusion (MATE) family, [210]. *L. monocytogenes* is exposed to many acids, such as propionic, benzoic, salicylic, and lactic, which are commonly used as food preservatives and disinfectants. There are four potential mechanisms responsible for the homeostasis maintenance in *L. monocytogenes* such as acid tolerance response (ATR), glutamate decarboxylase (GAD), arginine deiminase (ADI), and F1F0-ATPase activated to protect the cell against lethal action while GAD (gadD1, gadD2, gadD3, gadT1, and gadT2), which elevates the cell pH for survival [211].

Biofilms act as a bacterial protection layer against disinfectant agents and antibiotics and enhance the removal of metabolites and the transfer of nutrients. The formation of biofilm is not uniform but continuous and is regulated by quorum sensing [212]. In Mexico, where a total of 17 *L. monocytogenes* strains were collected from cilantro (*Coriandrum sativum*) lettuce (*Lactuca sativa*), and broccoli (*Brassica oleracea var. italica*), the isolates of *L. monocytogenes* belonged to phylogenetic groups I.1 (29.4% (5/17); serotype 1/2a) and II.2 (70.5% (12/17); serotype 1/2b); strains containing *Listeria* pathogenicity islands (LIPIs) were also identified at prevalence rates of 100% for LIPI-1 and LIPI-2 (17/17), 29.4% for LIPI-3 (5/17), and 11.7% for LIPI-4 (2/17) [213]. Phenotypic tests showed that 58.8% (10/17) of cadmium-resistant *L. monocytogenes* isolates had co-resistance to BC at 23.5% (4/17), and all *L. monocytogenes* strains exhibited moderate biofilm production. Moreover, the genes *Ide*, *tetM*, and *msrA*, associated with efflux pumps *Lde*, tetracycline, and ciprofloxacin resistance, were detected at 52.9% (9/17), 29.4% (5/17), and 17.6% (3/17), respectively. Communication among bacteria is predominantly via signalling molecules, the autoinducers, which are responsible for virulence factors, bacterial gene expression, and other wide-ranging functions through quorum sensing [214]. Two quorum-sensing systems, *lux* and *agr*, have been identified in *L. monocytogenes* [215]. Studies from the United States, Portugal, and the Netherlands reported that these systems, OpuC (*opuCABCD*), Gbu (*gbuABC*) and BetL (*betL*), and others, facilitate the accumulation of compatible solutes under high osmolarity conditions, and they facilitate the transportation of carnitine and glycine betaine into the cytoplasm from outside of the cell membrane to combat high salt stress [216,217]. To overcome harsh and changing conditions, gene expression in *Listeria* is tightly regulated by transcriptional regulator systems, including sigma (σ) factors, transcriptional activators, and repressors [218]. The resilience of biofilms is attributed to extracellular polymeric substances (EPS), which include lectins, proteins, nucleic acids (eDNA), exopolysaccharides, and lipopolysaccharides [215]. The *actA* gene, present in ST_120_, plays an essential role in the initial step of biofilm formation by regulating motility, along with other intrinsic and extrinsic factors such as nutrients [166]. Another study highlights hemolysin in *Listeria* as a function related to biofilm formation [219].

### 3.7. Foodborne Cases and Foodborne Incidence

The severity of listeriosis depends not only on the infectious dose and the age and immune status of the patient but also on the virulence of the ingested strain or subtype, the expression of key virulence genes, and other factors. Temperature is one of the most important environmental parameters affecting both food quality and food safety. Consumers suffer from foodborne illness due to consuming produce contaminated with extremely low levels of toxins [220]. Several studies reported cases of listeriosis and the number of deaths due to poor agronomic practices. The transcription of virulence genes can be affected by the storage temperature of fresh produce, suggesting the transcriptomic response and virulence of the pathogen to be complex [221]. The United States reported 10 cases of listeriosis from ready-to-eat celery (*Apium graveolens* L.), with five deaths, and 99 cases of romaine lettuce listeriosis, with 15 deaths [222]. There were 19 cases of listeriosis in Switzerland, with only one death, due to *L. monocytogenes* contamination of leafy green vegetables [223]. A review in South Africa highlighted that *Internalin B* (*InlB*) is a protein that plays a major role in *L. monocytogenes* binding to enterocytes and the subsequent invasion of the intestinal barrier, in line with stressors that protect the pathogen from bile in the gastrointestinal tract and subsequent invasion [155]. Another outbreak associated with romaine lettuce was recorded across 19 states in the United States, where 84 people became ill, and 15 died. The Food and Drug Administration (FDA) tested random samples from True Leaf Farms in California, and the results were positive for *L. monocytogenes*. Approximately 30,000 pounds of chopped and bagged romaine lettuce, packaged in 90 cartons, was recalled [224].

The effective expression of these adhesion proteins and virulence genes under osmotic and acid stress conditions may contribute to a high infection potential. On average, the United States had 57 outbreaks of foodborne illness from fresh produce contamination each year, while foodborne outbreaks related to fresh produce in Japan declined by 33% between 2002 and 2012 [225]. New Zealand now reports three times as many fresh produce-linked outbreaks as in 2002 [225]. It is reported that foodborne diseases are under-reported and poorly investigated in South Africa due to a lack of surveillance and integrated management [226]. The study reported foodborne disease outbreaks from the NICD unit from 2013 to 2017 and concluded that there has not been a noticeable improvement in the notification and investigation of these outbreaks since then, and that the process remains the same. Delays in responding to outbreaks, including the use of appropriate testing tools, contribute to under-reporting of foodborne diseases in South Africa [227]. Furthermore, it is highlighted that there is an inconsistency in foodborne disease outbreak investigation and reporting at the local and district levels in South Africa, including a lack of epidemiological data. For example, many provinces in South Africa reported few to no foodborne outbreaks over the five years, including the Northern Cape (0.3%), Gauteng (19%), and Mpumalanga (12%) [226].

Few studies from Marseille and Paris (France) reported *L. ivanovii*-associated gastroenteritis and bacteraemia in humans [1,31,32]. It is reported that food poisoning cases are under-reported and poorly investigated in South Africa due to a poor surveillance system and poor integrated management. From a practical point of view, understanding food safety and the mechanisms of survival for *Listeria* is imperative. Several policies have been drafted, but there is no integrated system to curb foodborne diseases, monitor outbreaks, or prevent disease spread, indicating the necessity of a robust surveillance system in South Africa. Major challenges in combating *L. monocytogenes* include poor planning, poor application of food safety systems, and the persistence of *L. monocytogenes* in food. Thus, it is crucial for a comprehensive study to evaluate trends in pathogenic potential, aetiology across regions, foodborne disease, and mortality, including ecology, prevalence, and virulence determinants of pathogenic *Listeria*.

The primary implication of *L. ivanovii* might be greater than currently under-documented, as it is overshadowed by *L. monocytogenes* owing to the sporadic nature of cases comparing the two pathogens. Few studies now shed light on *L. ivanovii* in food products as an emerging and recognised foodborne pathogen. However, its host range was thought to be limited to ruminants, and its relative severity compared with *L. monocytogenes* makes human cases less common. The studies have begun to unpack the virulence genes of *L. ivanovii* compared to *L. monocytogenes* and have highlighted observed similarities, including its low virulence but potentially fatal outcome. Another distinguishable reason contributing to under-documentation is the lack of severe symptoms, which may be clinically less recognisable. Its lower virulence compared to *L. monocytogenes* means that, in a healthy person, it is asymptomatic, resulting in undiagnosed and unreported cases. There is less information available in the literature on the prevalence and distribution of *L. ivanovii* along the food chain. However, it appears that, apart from *L. monocytogenes*, *L. ivanovii* is the most frequently isolated *Listeria* species.

## 4. Conclusions

In most cases, small-scale farmers do not possess good agricultural practice certificates, which qualify them for broader knowledge of the farm-to-fork continuum. From a practical point of view, understanding food safety and the mechanism of survival for *Listeria* is imperative. Risk mitigation for *L. monocytogenes* and *L. ivanovii* should be integrated methods that aim to control the food chain supply continuum from farm to market establishment. Poor hygiene and inadequate manufacturing processes increase the risk of contamination in processing facilities and foods. Food safety awareness and health education are essential for local small-scale farmers and markets, as they help mitigate food contamination and understand microbial infections. There is a need to develop more robust microbial monitoring to assess the safety of irrigation water and to regularly monitor soil microbial status, including fertiliser management for leafy green growth. It is imperative to know when manure should be applied as a fertiliser, and the harvest time is critical to avoid crop contamination. It is also important that operating procedures be strictly followed, particularly regarding hand-washing facilities, to prevent contamination, as these pathogens are present in environments with poor hygiene and faecal contamination. Revision of hygiene and sanitation standards, along with frequent monitoring, is required to prevent negligent personnel hygiene practices, recalls, and facility shutdowns due to outbreaks.

Ultimately, the role of governments and public health institutions in controlling and regulating the primary food sector and in providing effective consumer risk education will be crucial in preventing large outbreaks in the future. In addition, food safety regulations that include standards and limits for *L. monocytogenes* and *L. ivanovii* in leafy greens will be necessary to provide consumer protection and improve understanding of the *Listeria* spp. genetic and virulence factors, underlying strategies and mechanisms (including their ecology), and the pathogen’s ecology are key to developing novel treatments to control this pathogen, including addressing gaps such as infection dose and regulations. In most African countries, if not all of them, listeriosis receives little attention in public health systems.

## Figures and Tables

**Figure 1 microorganisms-14-00076-f001:**
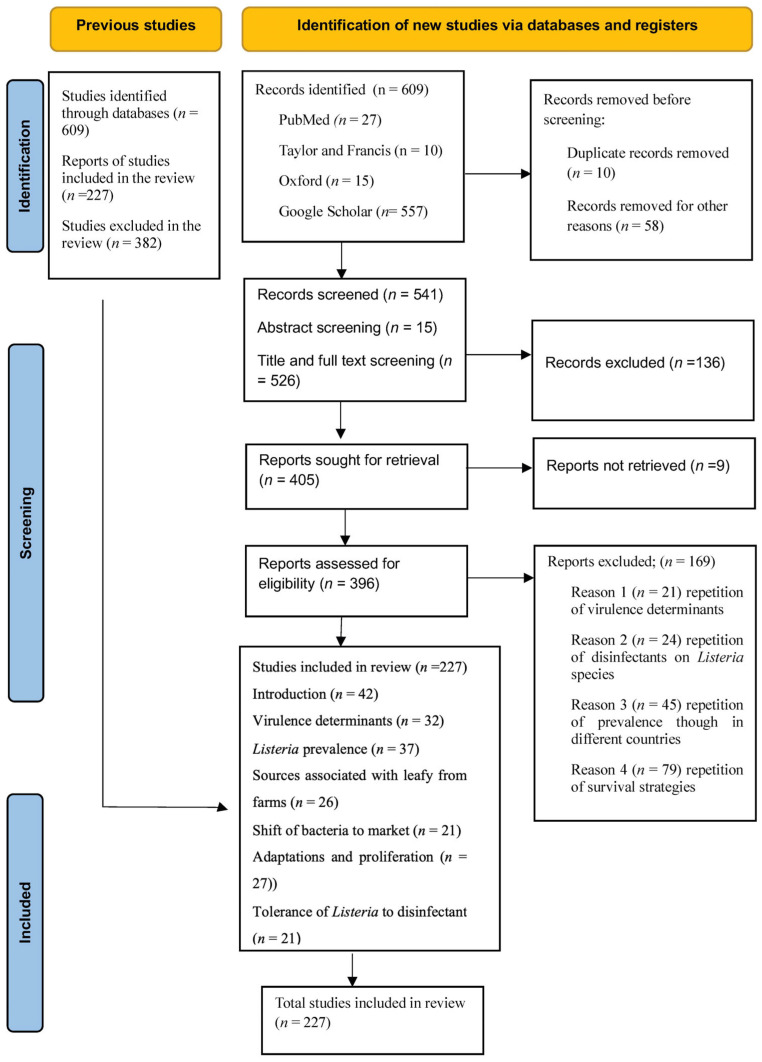
Flow diagram illustrating eligible studies that were identified during the selection process by following the PRISMA guidelines.

**Table 1 microorganisms-14-00076-t001:** *L. monocytogenes* virulence genes were detected in various leafy green vegetables from various countries, including the methods [66].

Country	Year	Leafy Greens	Virulence Genes	Method Detected	References
Florida and Greater Washington, DC	2002 and 2003	Conventionally grown fresh produce and two from organically grown fresh produce	4b, 4d, 4e (*n* = 11; 42.3%)	PCR genotyping using pulsed-field gel electrophoresis	[61]
São Paulo, Brazil	April and August 2008	Salad mixes and lettuces	*inlA* gene (*n* = 16; *nlC* and *inlJ* genes (*n* = 15; 97%)	Multiplex PCR	[48]
US and Canada	2015–2016	Packaged leafy salad (spinach)	4b, ST382		[67]
China	2011–2016	Vegetables, including lettuce and coriander	*prfA*, *mpl*, *plcA*, *inlB*, *plcA*, *hly*, *iap*, and *actA* genes (*n* = 23), *llsX* gene (*n* = 3; ST1 and ST3); *ptsA* gene (*n* = 11) ST87	PCR	[8]
Chile	2020	Leafy vegetable salads	*hlyA*, *prfA*, and *inlA* were detected by; 7 strains were 1/2a serotype, and one was a 4b strain	PCR	[68]
South Africa	2019	Cabbage, spinach	*inlA*, *inlC*, *prfA*, *plcA*, *hly*, *plcB (n* = 103; 100%); *inlJ* (*n* = 91; 88%); *inlB* (*n* = 89; 86%); *mpl* (*n* = 94; 92%); *actA* (*n* = 87; 85%); LIPI-1 (*n* = 77; 75%)	PCR	[40]
Mazandaran and Golestan provinces, northern Iran.	During July 2018 and January 2020	Cabbage, spinach, lettuce, parsley, coriander and dill	1/2a and 4b; including *hlyA*, *plc*, *iap*, and *actA*	PCR	[69]
Beijing, China	August 2019 to April 2021	Vegetables including cabbage	*inlA* and *inlB* and LIPI-1 pathogenic islands (*prfA*, *plcA*, *hly* and *actA*)	WGS MLST	[70]
Michigan and West Virginia	Summer 2019	Romaine lettuce, collards, celery, basil, and kale	CC1 (ST1) and CC4 (ST219) of lineage I; CC7 (ST7) and CC11 (ST451) of lineage II. CC4 and CC7 were present in the romaine lettuce sample. CC1. LIPI-1 and LIPI-3, CC4 contained LIPI-1, LIPI-3, and LIPI-4. CC7 and CC11 had LIPI-1	PCR and genotyping by pulsed-field gel electrophoresis (PFGE). MLST and core-genome multi-locus sequence typing (cgMLST)	[71]
South Africa	2018	Spinach	LIPI I, 1/2b, ST5	WGS	[72]
Suleimani and Halabja, Iraq	1 December 2024, 31 May 2025	Lettuce, cauliflower, cabbage, spinach, celery	*prfA* and *inlA* genes were each detected in 41.6% of isolates, and *hlyA* in 33.3%.	PCR	[73]
South Africa	2019	Cilantro/Coriander	LIPI I, 1/2b, ST5	WGS	[72]

**Table 2 microorganisms-14-00076-t002:** *Listeria* spp. associated with foodborne disease and its minimum infective dose [87,88].

*Bacteria* sp.	Disease Type	Tolerated Dose	Source of Food
*L. monocytogenes*	Infection (invasive) Infection Non-invasive (febrile gastroenteritis)	Risk individuals (10^5^ to 10^7^ CFU; 0.1 to 10 million CFU) Healthy people (10^7^ to 10^9^ CFU; 10 to 100 million CFU) 10^6^ CFU required to cause febrile gastroenteritis	Leafy green vegetables; ready-to-eat salads Leafy green vegetables; ready-to-eat salads Leafy green vegetables; ready-to-eat salads
*L. ivanovii*	Infection	Not specified	Leafy greens; ready-to-eat salads

**Table 3 microorganisms-14-00076-t003:** *Listeria monocytogenes* and *Listeria ivanovii* are sources of contamination from leafy green vegetables collected from farms and markets.

Area	Sampling Area	Contaminated Commodities	Epidemiological Sources (Sources of Contamination)	Causative Pathogen	References
Texas	Cabbage farms including packing sheds and (*n* = 6)	Cabbage (*n* = 425), water (*n* = 205), and environmental (*n* = 225)	Packing sheds surfaces	*Listeria monocytogenes*, *Listeria ivanovii*	[130]
Southern United State	Farms (*n* = 13) and packing sheds (*n* = 5)	Vegetables (leafy greens including herbs (*n* = 398)	Pre-harvest and post-harvest contamination	*Listeria* *monocytogenes*	[131]
Norwegian	Collected from 12 fresh farm producers	179 samples of organically grown lettuce were positive for *L. monocytogenes*	Irrigation water may be a point source of *L. monocytogenes* contamination.	*Listeria* *monocytogenes*	[132]
New York	Produce farms (*n* = 21)	Fields (*n* = 263) and environmental samples (soil, including water *n* = 600)	Chlorine washing may have not decreased microbial load	*Listeria* *monocytogenes* (Lineages I, II, and IIIa)	[133]
Turkey	A total of 164 leafy green vegetable samples were collected from various agricultural fields	14 samples (3 basils, 1 dill, 1 garden cress, 2 kales, 1 lettuce, 1 mint, 2 parsleys, 1 purslane and 2 rockets) were positive for *L. monocytogenes*	Soil and improper hygiene during processing	*Listeria monocytogenes*	[134]
Osijek, Croatia	Fresh produce markets	Lettuce and cabbage	Cold chain abruption	*Listeria ivanovii*	[135]
New York	Produce farms (*n* = 5)	Soil, water faeces, and drag swabs (*n* = 588)	Water, roads and urban development, and pasture/hay grass) influenced the likelihood of detecting *L. monocytogenes.*	*Listeria monocytogenes*	[136]
South and North Carolina, Georgia, Kentucky, California, Tennessee	Produce Farms	Organic fertilisers (*n* = 103)	Organic fertilisers	*Listeria monocytogenes*	[137]
Kaduna State, Nigeria	Collected from markets	Coleslaw, cabbage and lettuce (*n* = 335)	Poor washing methods	*Listeria monocytogenes*	[138]
North-Western Nigeria	Fresh produce markets (336 samples)	Cabbage (*n* = 34), lettuce (*n* = 48)	Irrigation water and soil	*Listeria ivanovii*	[139]
South-Western Nigeria	Fresh produce markets	Cabbage and lettuce (*n* = 555)	Poor agricultural practises	*Listeria monocytogenes*	[140]
South Africa	Vegetable farms (*n* = 4), small-scale farm (*n* = 1), and homestead gardens (*n* = 40)	A total of 474 samples comprising cabbage (*n* = 334), baby spinach (*n* = 84) and lettuce (*n* = 56)	Irrigation water (commercial and small-scale farm, and homestead gardens)	*Listeria monocytogenes*	[141]
Maryland	Organic farms (*n* = 7)	Produce (tomatoes, leafy greens, peppers, cucumbers including water, and surface water (*n* = 206)	Washed leafy greens carried higher levels of some microbial indicators, possibly because of the lack of sanitiser in the wash water.	*Listeria monocytogenes*	[142]
New York	Spinach Field (*n* = 2)	1092 soil, 334 leaf, 14 faecal, and 52 water (*n* = 1492)	Irrigation water may be a point source of *L. monocytogenes* contamination.	*Listeria monocytogenes* sig*B* allele	[84]
South Africa	Spinach (*n* = 4) and cabbage farm (*n* = 5)	Raw spinach phyllosphere (*n* = 60) and cabbage (*n* = 75)	Livestock manure possible contaminant and improper hygiene during processing from primary production	*Listeria monocytogenes*, *Listeria ivanovii*	[66]
Spain	Leafy greens (*n* = 483)	Leafy greens including lettuce	Water circulation system and soil contamination	*Listeria monocytogenes*	[143]
Ireland	Leafy greens (*n* = 160)	Spinach, rocket, and kale produce	Harvesting conditions influenced *L. monocytogenes* growth conditions	*Listeria monocytogenes*	[144]

## Data Availability

The original contributions presented in this study are included in the article. Further inquiries can be directed to the corresponding author.

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
