# Peer review of "Listeria monocytogenes and Listeria ivanovii Virulence and Adaptations Associated with Leafy Vegetables from Small-Scale Farm and a Shift of Microbiota to a New Niche at Markets: A Systematic Review"

_microorganisms, 2025, doi:10.3390/microorganisms14010076_

Round 1
Reviewer 1 Report
Comments and Suggestions for Authors
The scope of the paper is unclear to me. From the title it is not clear what the focus is of the paper and also from the introduction it is not clear. apparently the main focus should be leafy vegetables and new niches at markets (the second is unclear to me what is meant). However, the points addressed are not focused - at many points the scope is too wide. for instance the virulence factors are not focused to the scope of the title, but are at different points wider addressed then the scope of the title.
The methods are described very unclear. The description is not concrete "including but not limited to", which makes it unclear what was done exactly. moreover the process is described in different paragraphs, but actually each paragraph seems to describe the same part of the process. But some other parts are not described at all. exclusion criteria are for instance not clear.
Description of the results are not matching up. how to go from 384 to 382 is unclear. and how do they next go back up to 393?
discussion is not focused enough. should be much more focused with regards to the title.
Also sentence that follow each other seem to be in random order, jumping from one topic to another. For instance, in line 454 to 455 why suddenly jumping from attack rates and incubation periods to survival in the environment. If there was a reasoning behind this it is not clear from the text. this needs to be added for readability. this type of topic changes occur a lot. But the text is already very lengthy, so when adding text for readability also a lot of data needs to be cut.
It seems that the content per paragraph is not related to the title of the paragraph. this needs be put more in line with each other.
For one reference that was checked, it appears that the exact text in the manuscript was copied from the abstract (Iwu et al.). it was not checked for others, but already for one this is not how science is done. Also in a review the authors need to write their own texts. Also the conclusion from the Iwu paper are not in line with results with other papers, so some discussion need to added for such results and not just copy the results.
the conclusion is also not in line with the scope of the title of the manuscript. there is nothing mentioned about leafy greens for instance.
Comments on the Quality of English Language
The quality of the English language is not good enough. The level is actually such that the manuscript is hard to comprehend. It needs a lot of rewriting.
Author Response
the comments has been sent as attachment
Reviewer 2 Report
Comments and Suggestions for Authors
The presented article includes an overview of the major pathogenicity factors of Listeria. The article is well-written in terms of depth of analysis of the research question.
However, to improve the perception of the review study, I recommend that the authors revise section 2, Materials and Methods. Such a section looks unusual in a "Review"-type article. It would be better to replace it with a "Research Methodology" section and briefly describe the main stages concisely.
Section 3, Results and Discussion, also seems odd, especially since there is a separate section 4, Discussion. It would be more logical to combine section 3 with section 2 under a single title.
Author Response
comments have been sent as word document below
Reviewer 3 Report
Comments and Suggestions for Authors
The review addresses a relevant and timely topic, especially regarding public health concerns from small-scale farm produce entering market chains, with a strong emphasis on South Africa and the global context. Overall, the approach is sound. However, the authors should address some issues to clarify and strengthen the manuscript. Here follows specific comments:
Abstract
Please split complex sentences and clarify the main findings and implications for food safety in the context of leafy greens from small farms and market environments.
Introduction
1) I suggest that the authors emphasize more on regional surveillance gaps and epidemiological data from South Africa
Methods and Search Strategy
1) Please include the rationale for choosing specific timeframes and the exclusion of studies focusing on other Listeria species.
2) Please include additional information on how redundancy was addressed in cases of reports focusing on single variables from different countries.
Results and Discussion
1) Please include a brief interpretive remark in tables and summary figures to highlight trends observed across regions and commodity types.
2) Provide greater emphasis on specific knowledge gaps, such as the under-documented role of L. ivanovii in outbreaks.
Conclusion
1) Summarize more directly the significant risk factors observed and propose mitigation strategies for Listeria contamination in leafy greens.
General comment:
Please revise throughout the manuscript to correct typographical errors.
Author Response
comments has been sent as an attachement
Round 2
Reviewer 2 Report
Comments and Suggestions for Authors
The paper can be accepting